# Evaluation of Selected Heavy Metals Contaminants in the Fruits and Leaves of Organic, Conventional and Wild Raspberry (*Rubus idaeus* L.)

**Marta Kotuła** [1], **Joanna Kapusta-Duch** [1,*] and **Sylwester Smoleń** [2]

1 Department of Human Nutrition and Dietetics, Faculty of Food Technology, University of Agriculture in Krakow, Al. Mickiewicza 21, 31-120 Krakow, Poland; marta.kotula@urk.edu.pl
2 Unit of Plant Nutrition, Institute of Plant Biology and Biotechnology, Faculty of Biotechnology and Horticulture, University of Agriculture in Krakow, Al. Mickiewicza 21, 31-120 Krakow, Poland; sylwester.smolen@urk.edu.pl
* Correspondence: joanna.kapusta-duch@urk.edu.pl; Tel.: +48-12-662-48-16

**Abstract:** Chemical pollutants contaminate the air, soil and then plants, which are the main source of xenobiotics for humans. Many consumers perceive that organically grown foods are of better quality, healthier and more nutritious than conventionally grown foods. The aim of the study was to analyse raspberry fruits and leaves from organic, conventional and wild crops in Poland for two years, independently of climatic and agrotechnical conditions. The metal contents (Al, B, Cu, Mn and Zn) were determined using the validated Inductively Coupled Plasma Optical Emission Spectrometry (ICP-OES) and by Inductively Coupled Plasma Mass Spectrometry (ICP-MS/MS) (As, Cd, Pb, Cr, Co, Mo, V, Sr, Sb, Ti and Tl). The raspberry fruits from organic farming contained more ($p \leq 0.05$) Cd, Zn, Mn and V compared to those from conventional cultivation. Fruits of wild-growing raspberry had more Cd, Zn, Co and Mn compared to other crops. Wild-growing raspberry leaves had more Cd, Pb, Zn, Mn and Tl than the other leaves. The raspberry leaves from organic cultivation had more Cr, Cd and Pb compared to leaves from conventional cultivation. The leaves and fruits of wild-growing raspberry are characterized by a significantly higher content of heavy metals.

**Keywords:** contaminants; cadmium; vanadium; lead; thallium; chromium; boron; strontium; titanium; raspberry

## 1. Introduction

Environmental pollution with heavy metals is a growing problem worldwide. Their detrimental effect on human and animal health includes, i.e., dysfunctions of the gastrointestinal tract, nervous and immune systems or the formation of neoplasms [1]. Heavy metals such as chromium, zinc, molybdenum, cobalt, manganese and vanadium, in small amounts, are essential for the proper functioning of the human body. On the other hand, the intake of elements such as arsenic, cadmium, copper, chromium and lead is toxic to humans and may have harmful effects on health [2].

Natural and anthropogenic activities contribute most to the heavy metal accumulation in the environment. The main heavy metal sources include sewage sludge, irrigation with purified sewage, atmospheric pollution as well as mining, milling, fossil fuel combustion and agrochemicals [3,4]. Food production in contaminated areas leads to the accumulation of these substances in vegetables and fruit. This, in turn, negatively affects food safety and human and animal health, as well as the quality and growth of crops [2].

Conventional agriculture uses fertilizers and agrochemicals to increase yields. The continued and unjustified excessive use of these can damage the environment and biodiversity and may affect human health, causing chronic health problems [5]. It has been proved that crops fertilization has the greatest impact on the environment [6]. The compounds

used, depending on their chemical structure, remain permanently in the environment and can accumulate in fruits, vegetables and also in animal meat. In addition, some of these compounds are dangerous already at low concentrations [7]. The accumulation of heavy metals in the plant takes place mainly through the roots and, to a lesser extent, through the leaves [4].

Organic farming is based on a diversified farming system that includes the use of organic fertilizers and crop rotations that preserve and support the ecosystem [8]. According to Regulation (EU) 2018/848 of the European Parliament and of the Council of 30 May 2018, organic farming is an agricultural method of producing food without the use of non-natural substances and processes [9]. This protects the environment, the climate and natural resources, as well as biodiversity.

The latest literature investigates the use of nanomaterials lowering the content of heavy metals in the plant. This effect would be achieved by reducing the heavy metal content in soil, regulating the expression of genes responsible for the transport of these substances in plants and by enhancing the capacity of barriers for capturing heavy metals [10–12].

Poland is the largest raspberry producer in the European Union. In 2020, the raspberry harvest reached 116,000 tonnes, an increase of 53% over 2019 [13]. Raspberries are a rich source of antioxidant components, such as polyphenols (flavonoids with anthocyanins and phenolic acids), carotenoid, vitamin C and micro- and macroelements [14].

The scientific research on the properties of raspberry (*Rubus idaeus* L.) confirms the preventive effect of bioactive components in the fight against non-communicable diseases, such as obesity, diabetes, cancer, hypertension and other cardiovascular diseases, which are currently serious problems in society [14,15]. Raspberry fruits are characterised by antioxidant, anti-inflammatory, anticancer, antiviral and antimicrobial properties [16–18]. In addition, the significant role of raspberry fruit and leaves in protecting against the development of diabetes is currently pointed out [15,19]. In the light of current knowledge, raspberry ketone is being studied in the context of the treatment of obesity. There is no proven mechanism of action of this compound in the fight against obesity, but its beneficial effects against diabetes and protection of the heart and liver are noted [20,21]. In addition, raspberry leaves have been used for centuries in women during pregnancy and childbirth to relax the uterus, support contractions and prevent haemorrhages. However, more human studies are needed to document their effectiveness and safety [22].

Fruits and vegetables accumulate heavy metals from the soil as they grow [8]. These can be transferred through the food chain due to their non-biodegradability. Moreover, their mere presence is already a threat to human health, although some do not pose a threat until they are present in increased quantities. Zinc, copper, chromium, molybdenum, cobalt and vanadium are elements necessary for the proper body functioning and therefore become toxic only at higher concentrations. The intake of heavy metals in the diet can have negative consequences. Their excessive intake, depending on the amount, time and level of exposure, is toxic and causes many chronic diseases.

The main source of heavy metals in plants is the growing environment. Contamination can also result from the migration of elements from the packaging or technological processes; however, in the case of fruits, which are mainly eaten raw, this phenomenon does not occur frequently. The contamination of fruit and vegetables is caused by polluted water used to irrigate plants, fertilisers, soil pollutants and anthropogenic effects such as industrial emissions and household pollution [23]. Fruits and vegetables should be the base for a healthy diet. According to the recommendations of the Polish Institute of Food and Nutrition, the daily serving of fruit and vegetables should be at least 400 g; however, higher vegetable consumption is recommended. The same recommendation was given by the Food and Agriculture Organization of the United Nations of the World Health Organization (FAO/WHO). Rusin et al. [23] and Antić-Mladenović et al. [24] found that the Pb and Cd contents of lead and cadmium in vegetables and fruits was exceeded only in a few cases, in 9.3% of vegetables, 1% of fruits (Pb) and in 2% of vegetables (Cd). This may,

however, pose the exposure to heavy metals and some risk to consumers, especially since vegetables and fruits should be an important part of the daily diet.

According to the available literature, data on the content and identification of heavy metals in the fruits and leaves of raspberry (*Rubus idaeus* L.) have been already published, but there are no publications with such a wide range of elements. It is also difficult to find papers comparing the organically and conventionally grown fruits and leaves of raspberries with the wild types of raspberries, which are a viable option for the consumer and are not yet fully explored in the available data. Therefore, the aim of the study was to analyse raspberry fruits and leaves from organic, conventional and wild crops in Poland for two years. The content of heavy metals, i.e., arsenic (As), cadmium (Cd), lead (Pb), aluminium (Al), chromium (Cr), zinc (Zn), cobalt (Co), molybdenum (Mo), copper (Cu), manganese (Mn), vanadium (V), boron (B), strontium (Sr), antimony (Sb), titanium (Ti) and thallium (Tl), was determined.

With a view to this goal, the research hypothesis tested was that the selected heavy metals content in raspberry fruits and leaves grown wild, those from organic farms and those randomly purchased at local markets (conventional) differed statistically significantly, independently of the climatic and agrotechnical conditions.

The implementation of such a scheme according to the research experience, which includes this work, allows new/innovative treatment of the problem, and the obtained conclusions give clear recommendations about the choice of fruits for the potential consumer, depending on the source, but it is not for the farmers who are obliged to modify the parameters of the cultivation.

## 2. Materials and Methods

### 2.1. Plant Material

The experimental material consisted of fruits and leaves of the raspberry (*Rubus idaeus* L.) obtained from conventional and organic cultivations as well originating from wild-growing plants. Organic and conventional raspberries were of the *Polana* variety. Samples were collected in August 2020 and 2021. In this aim, the study was conducted for 2 years because only the average value of the various evaluated parameters of the successive iterations obtained in such a long period of time may be an objective value that is representative independently on the resultant influence of all interfering factors, climatic and agro-meteorological. The conventional raspberry was produced in Jadowniki (Brzesko district, the Małopolskie Voivodeship). The organic raspberry came from a certified farm located in Bębło, near Kraków (Małopolskie Voivodeship), while wild-growing raspberry was collected in the forests near Siedlec (a short distance from Częstochowa, Śląskie Voivodeship).

### 2.2. Soils

The soils of forests near the village of Siedlec are classified as pseudopodsolic and brown soils, according to The World Reference Base (WRB) Podzols and Endocalcaric Cambisols, respectively. The majority of soils in the commune are made of Quaternary deposits, i.e., sand, gravel and peat, as well as limestone. These are the areas of the Polish Jurassic Highland, which is a recreational place with a large number of forests and green areas. There are mineral deposits (moulding sands and limestones) in this place. There is no industry in these areas, and the source of the presence of heavy metals may be a high degree of danger to groundwater.

The location of ecologically grown raspberries is characterized by Eutric Cambisols and Luvisols by the system of WRB. These areas occur in the vicinity of Krakow Valleys, also belonging to the Polish Jurassic Highland. However, this area is located near the road leading to Krakow, which, due to increased traffic, may be a source of pollution by car exhaust fumes.

Forest soils in the village of Jadowniki are classified as leached brown and acidic brown soils, the Eutric Cambisols and Dystric Cambisols, respectively. Most of the soils in the municipality are made up of five-door sediments. They were formed on loess, loess loams,

morainic clays, gravels, sands and river silts. These lands are part of the Western Carpathian Foothills region. These areas are close to industry and roads leading to Kraków, Tarnów and Rzeszów, which, due to increased traffic, can be a source of automobile exhaust pollution.

The research was carried out independently of climatic and agrotechnical conditions. The research was of a more consumer nature in order to ascertain differences between the fruits and leaves of raspberries from more or less controlled cultivation. The experimental material was harvested at the maturity stage. In pre-treatment, freshly harvested fruits and raspberry leaves were sorted and cleaned. The samples were then frozen on plastic plates at $-80\,^\circ$C, freeze-dried using a Christ Alpha 1–4 freeze drier and ground in a grinder (Tecator Foss, Hillerød, Sweden).

*2.3. Determination of Selected Heavy Metals*

The contents of micronutrients and trace elements were measured in air-dried samples. The 0.5 g samples were placed into 55-mL TFM vessels and were mineralised in 10 mL 65% super pure $HNO_3$ (Merck no. 100443.2500) in a Mars 5 Xpress (CEM, Matthews, NC, USA) microwave digestion system. The following mineralisation procedure was applied: 15 min (the time needed to achieve a temperature of 200 $^\circ$C) and 20 min maintaining the temperature. After cooling, the samples were quantitatively transferred to 25-mL graduated flasks with redistilled water. Contents of Al, B, Cu, Mn and Zn were determined using a high-dispersion inductively coupled plasma optical emission spectrometer (ICP-OES, Prodigy Teledyne Leeman Labs, Mason, OH, USA) [25]. Liquid samples were injected into a radiofrequency-(RF) induced argon plasma. The sample mist reaching the plasma was rapidly dried, vaporized and excited by collisions at high temperature. Atomic emission emanating from the plasma were viewed in a radial or axial configuration, collected with a lens and imaged at the instrument's entrance slit. Due to the lower compactness of As, Co, Cr, Cd, Mo, Pb, Sb, Sr, Ti, Tl and V, the analysis of these elements was performed by inductively coupled plasma mass spectrometry (ICP-MS/MS) with a triple quadruple spectrometer (iCAP TQ ICP-MS ThermoFisher Scientific, Bremen, Germany). A quadrupole consists of four parallel hyperbolic or cylindrical metallic rods positioned in a square array. Radio frequency alternating current (AC) and direct current (DC) potentials are applied to the rods, creating a time-varying electric field in the centre through which ions pass.

*2.4. Statistical Analysis*

All measurements were performed in at least triplicate, and all data were expressed as mean $\pm$ standard deviation (SD). All calculations were carried out using Statistica software v. 13.1 PL (Dell Inc., Tulsa, OK, USA). Significant differences in the mean values were compared using two-way ANOVA and Duncan's test with $\alpha = 0.05$. Cultivation and harvest time have been taken as two factors.

## 3. Results and Discussion

*3.1. Heavy Metals Content in Raspberry Fruit*

The content of heavy metals (As, Cd, Pb, Al, Cr, Zn, Co, Mo, Cu, Mn, V, B, Sr, Sb, Ti, Tl) in raspberry fruits from conventional, organic and wild crop and two years of cultivation are presented in Table 1.

**Table 1.** The content of selected heavy metals in raspberry fruits from organic, conventional and wild plants (mg kg−1 dry matter).

| Heavy Metals | Cultivation | Conventional | | Organic | | Wild Crop | |
|---|---|---|---|---|---|---|---|
| | Year | 2020 | 2021 | 2020 | 2021 | 2020 | 2021 |
| Arsenic (As) | | 0.40 [b,c] ± 0.03 | 0.03 [a] ± 0.00 | 0.43 [c] ± 0.01 | 0.03 [a] ± 0.00 | 0.40 [b] ± 0.01 | 0.04 [a] ± 0.00 |
| Cadmium (Cd) | | 0.04 [a] ± 0.00 | 0.04 [a] ± 0.00 | 0.05 [a] ± 0.00 | 0.19 [b] ± 0.01 | 1.00 [d] ± 0.04 | 0.60 [c] ± 0.02 |
| Lead (Pb) | | 0.04 [a] ± 0.02 | 0.09 [c] ± 0.01 | 0.06 [a,b] ± 0.01 | 0.10 [c] ± 0.00 | 0.07 [b] ± 0.00 | 0.16 [d] ± 0.01 |
| Aluminium (Al) | | nd | 6.49 [a] ± 1.11 | 6.10 [a] ± 3.14 | 8.21 [a] ± 2.79 | nd | 19.26 [a] ± 3.91 |
| Chromium (Cr) | | 3.76 [b,c] ± 0.15 | 0.10 [a] ± 0.26 | 3.91 [c] ± 0.56 | 0.03 [a] ± 1.29 | 3.60 [b] ± 1.22 | 0.09 [a] ± 2.17 |
| Zinc (Zn) | | 27.59 [a] ± 1.23 | 28.49 [a] ± 0.26 | 27.52 [a] ± 0.56 | 32.18 [b] ± 1.29 | 35.76 [c] ± 1.22 | 32.92 [b] ± 2.17 |
| Cobalt (Co) | | 2.08 [c] ± 0.11 | 0.02 [a] ± 0.00 | 2.05 [b,c] ± 0.02 | 0.06 [a] ± 0.01 | 1.95 [b] ± 0.12 | 0.03 [a] ± 0.00 |
| Molybdenum (Mo) | | 2.25 [a] ± 0.03 | 0.60 [c] ± 0.00 | 2.17 [a] ± 0.07 | 0.27 [b] ± 0.01 | 2.00 [e] ± 0.12 | 1.01 [d] ± 0.00 |
| Copper (Cu) | | 4.78 [a] ± 0.65 | 5.76 [b] ± 0.82 | 4.64 [a] ± 0.39 | 4.69 [a] ± 0.15 | 5.96 [b] ± 0.64 | 6.70 [b] ± 0.22 |
| Manganese (Mn) | | 29.00 [a] ± 1.00 | 24.21 [a] ± 0.53 | 28.42 [a] ± 0.79 | 93.44 [b] ± 1.41 | 340.70 [c] ± 20.58 | 100.90 [b] ± 1.61 |
| Vanadium (V) | | 0.02 [a,b] ± 0.00 | 0.01 [a] ± 0.00 | 0.04 [c] ± 0.01 | 0.02 [a,b] ± 0.00 | 0.03 [b] ± 0.00 | 0.10 [d] ± 0.01 |
| Boron (B) | | 12.83 [b] ± 2.69 | 10.36 [a,b] ± 1.02 | 11.13 [a,b] ± 1.72 | 9.85 [a] ± 0.72 | 8.44 [a,c] ± 0.35 | 6.93 [c] ± 0.25 |
| Strontium (Sr) | | 4.10 [b] ± 0.12 | 4.96 [c] ± 0.23 | 3.03 [a] ± 0.43 | 4.44 [b] ± 0.20 | 3.03 [a] ± 0.13 | 5.37 [c] ± 0.25 |
| Antimony (Sb) | | 0.36 [b,c] ± 0.02 | 0.005 [a] ± 0.00 | 0.38 [c] ± 0.01 | 0.004 [a] ± 0.00 | 0.36 [b] ± 0.02 | 0.008 [a] ± 0.00 |
| Titanium (Ti) | | 0.88 [c] ± 0.15 | 1.49 [a,b] ± 0.07 | 1.91 [b] ± 0.10 | 1.66 [a,b] ± 0.28 | 1.3 [a,c] ± 0.16 | 2.83 [d] ± 0.58 |
| Thallium (Tl) | | nd | 0.00092 [a] ± 0.00026 | nd | 0.00051 [a] ± 0.00022 | 0.00263 [a] ± 0.00083 | 0.00777 [a] ± 0.00044 |

Results are presented as mean value ± standard deviation (n = 3). a, b, c . . . —means with different superscript letters in common differ significantly ($p < 0.05$), nd—not detected.

The result of the study showed that the concentrations of all heavy metals were among the analysed fruit samples without aluminium and thallium. The heavy metals content in the analysed samples was varied.

In the second year, the levels of arsenic (As), chromium (Cr), cobalt (Co), molybdenum (Mo) and antimony (Sb) were significantly lower in all examined cultures compared to 2020. In addition, there were considerable and statistically significant differences in the content of lead (Pb) and titanium (Ti) between the harvest years of conventionally grown raspberries. Furthermore, only the level of lead was higher in 2021. In organic raspberries, considerable and statistically significant differences were found between the contents of cadmium (Cd), manganese (Mn) and vanadium (V) depending on the harvest year. Most of the elements determined in wild-growing raspberry showed statistical significance compared to two years of cultivation, except for copper (Cu) and boron (B). This means that the compounds accumulation in the fruit did not vary significantly in the successive cultivation year. The wild-growing raspberry fruits were characterised by significant contents of cadmium, aluminium and manganese. More importantly, this occurred in both harvest years and was statistically significant compared to the remaining crops. In the second year of the study, less rainfall was observed, and consequently, there was less sorption of mineral nutrients by the plants. The presence of heavy metals in a plant depends on many factors, including the total and accessible content of these minerals in the soil, soil pH and organic matter content. In addition, the type and species of plant also have an influence on the uptake of heavy metals [26,27].

The arsenic content varied significantly between the examined years, the results being similar and independent of the cultivation type. In the available literature there are no studies on the As level in raspberries. Jedynak et al. [28] investigated its quantities in raspberries growing on arsenic-contaminated land. The As content in *Rubus idaeus* raspberry was 1.3 mg kg$^{-1}$ DW (dry weight). However, our results were much lower.

It may be due to the fact that the raspberries examined grew in three different areas characterised by insignificant arsenic content in the soil. According to the Food Standards Agency report, the content of this element in commercially available fruit and vegetables grown in As- enriched areas, and those devoid of this element was 0.0087 mg kg$^{-1}$ FW (fresh weight) and 0.0026 mg kg$^{-1}$ FW, respectively [29]. In this study, the average As content for conventional crops in 2020 and in 2021, calculated per fresh weight, was 0.067 mg kg$^{-1}$ and 0.004 mg kg$^{-1}$, respectively.

The chromium content of the analysed samples varied enormously depending on the year of cultivation and ranged from 3.91–3.76 mg kg$^{-1}$ in 2020 to 0.03–0.10 mg kg$^{-1}$ in 2021. In the second year of cultivation, the results were mutually insignificant; however, statistical significance was observed when compared to first year results. In 2021, the chromium content of organically grown fruits was statistically significant compared to wild-growing raspberry. Milinković et al. [8] reported various Cr amounts in the examined raspberries (*Rubus idaeus* L.) cv. *Willamette* and these values were not at the same level in the subsequent years of cultivation. According to Grembecka and Szefer [30] the Cr content in raspberry fruits was 0.76 mg kg$^{-1}$, while the mean value given by Antić-Mladenović [24] amounted to 0.54 mg kg$^{-1}$ DW for raspberries from different locations in Montenegro.

In the second harvest year, there were statistically significant differences in zinc content in raspberries from organic farming and wild-growing ones. In those from organic farming, the Zn content increased markedly (from 27.52 to 32.18 mg kg$^{-1}$ DW), while in wild-growing raspberries it decreased (from 35.76 to 32.92 mg kg$^{-1}$ DW). The results obtained for Zn content by other authors were lower [8,31,32] or higher [33], compared to our findings. In addition, the authors report slightly lower zinc contents in raspberry fruits (22.93, 25.34 and 21.63 mg kg$^{-1}$ DW depending on the type of soil [34] and slightly higher amounting to 33.6 mg kg$^{-1}$ [30]. In turn, Castilho Maro et al. [35], who examined various raspberry varieties, determined average Zn content in red *Rubus Idaeus* raspberries at the level of 23.55 mg kg$^{-1}$ DW, which is lower than in this study. Other authors also obtained a similar zinc content (3.7 mg kg$^{-1}$ FW), where in terms of FW, conventionally cultivated raspberry had 3.69 mg kg$^{-1}$ [36,37].

In comparison with the other examined raspberry fruits, the highest Mn content was recorded in wild-growing raspberry fruits (340.71 and 100.90 mg kg$^{-1}$ depending on the year of harvest). There was also a statistically significant increase in its content, by 328%, in organically grown raspberries. Lower levels were reported by other authors [8,34,35]. Manganese content stated by Grembecka and Szefer [30] in *Rubus idaeus* was similar to the value determined in the organic raspberry fruits harvested in 2021 and was 89.3 mg kg$^{-1}$. The Tolerable Upper Intake Levels (ULs) for manganese has been established by Institute of Medicine, Food and Nutrition Board at 11.0 mg for adults over the age of 19, regardless of gender. The content of this element in wild-growing raspberries harvested in 2020 and 2021 as well as in organic fruit from 2021 was too high, as it amounted to 64.51 mg, 18.99 mg and 13.03 mg kg$^{-1}$ when calculated per fresh weight. Such a high Mn content may have an adverse effect on human health.

The highest copper content, however, statistically insignificant, showed in wild-growing raspberry fruits. The results noted for the organic raspberries in two years of cultivation were close. In the fruits of conventional cultivation, a different content of this element was observed and increased significantly from 4.78 to 5.76 mg kg$^{-1}$ DW compared to the first year. Similar observations of copper content were made by other authors [8,34]. The results observed were also lower [31] or higher, amounting to 8.4 mg kg$^{-1}$ DW [30,33]. The Cu content determined by Castilho Maro et al. [35] in the fruits of the genus *Polana*, which is closest to the examined raspberries, was 5.1 mg kg$^{-1}$ DW.

Raspberry fruits cultivated in 2021 had a considerably higher Co content, which were 1.95–2.08 mg kg$^{-1}$ and 0.02–0.06 mg kg$^{-1}$ in 2020 and 2021, respectively. In 2020, similar contents of this element were noted in the raspberries from conventional and organic farming, while in wild-growing ones, the values were lower and statistically significant compared to the fruits conventionally grown. The Co content in Raspberry cv. *Willamette*

averaged 0.61 mg kg$^{-1}$ for fruit from conventional cultivation and 0.94 mg kg$^{-1}$ for fruit grown organically [8].

As for cadmium, its amount in organic raspberries increased by 380% compared to the previous harvest year. In the wild-growing raspberry, this value decreased from 1.00 mg to 0.60 mg kg$^{-1}$ DW. In addition, these results were statistically significant with each other, as well as compared with the other results (two years of conventional and organic raspberry cultivation in 2020). According to Rusin et al. [23], the Cd and Pb contents in raspberries (*Rubus idaeus* L.) were 0.116 mg kg$^{-1}$ and 0.111 mg kg$^{-1}$ DW. In the presented study, these fruits were purchased in a store, which may indicate that they were grown conventionally. When compared with our findings, the Cd and Pb content was lower (Cd contents were 0.05 and 0.04 mg kg$^{-1}$ and Pb contents 0.04 and 0.09 mg kg$^{-1}$ in the harvest years 2020 and 2021). In turn, Nejem et al. [33] found that the Pb content in the examined raspberries was significantly higher and amounted to 13.16 mg kg$^{-1}$ DW; however, they grew on contaminated soils. Higher Cd and Pb levels were also reported by other authors [31]. Sembratowicz et al. [38] investigated the level of cadmium and lead in raspberry fruit from the city centre and the outskirts. Lead content measured by the authors on the outskirts corresponded to our results, whereas cadmium content was lower.

The thallium content in the samples examined in 2020 was not determined for the raspberries from conventional and organic farming. Organic raspberries from the 2021 harvest had the lowest thallium content, while in wild-growing ones, its amount was the highest in both examined years; however, statistical significance was not found between the samples examined. According to Gruszecka-Kosowska [39], the thallium content in berries, which included raspberries, averaged 0.006 mg kg$^{-1}$ wet weight. This means that the fruits examined in this study had a lower amount of this heavy metal, since in wild-growing raspberry collected in 2021, containing the most thallium, its content was 0.0015 mg kg$^{-1}$ wet weight.

In the available literature, there are no data on the contents of aluminium, molybdenum, vanadium, boron, strontium, antimony and titanium in raspberry fruit. The highest Al content, however, statistically insignificant compared to other results, was found in wild-growing raspberry in 2021. As for Mo content, raspberry fruits harvested in 2020 contained more molybdenum than those collected in 2021.

Wild-growing raspberry is a fruit that is hardly available to consumers. Its selective occurrence and availability, which depends on weather conditions, contribute to this. Another obstacle is difficult harvest, as the fruits are smaller and the shrub is smaller than in organic or conventional cultivation. However, more and more shops offer organic and conventional raspberries. This study showed that organic raspberries contained significantly more Cd and Zn (harvest 2021) and vanadium (harvest 2020) compared to conventionally grown raspberries. The latter ones, cultivated in 2021, had higher statistically significant content of Mo, Cu and Sr. Of the aforementioned elements, cadmium poses a risk to human health, so with regard to the products containing its higher amounts caution is recommended. According to Commission Regulation (EU) No 488/2014 of 12 May 2014 [40], the tolerable weekly intake (TWI) of cadmium is 2.5 µg kg$^{-1}$ body weight. Per person weighing 70 kg, the maximum Cd dose should be less than 0.175 mg. Our results showed that organic raspberries harvested in 2021 and wild-growing raspberries collected in 2020 might pose a certain risk if consumed in large quantities. Calculating Cd content per fresh weight, wild-growing raspberry harvested in 2020 had the highest Cd content (0.109 mg kg$^{-1}$).

### 3.2. Heavy Metals Content in Raspberry Leaves

The content of heavy metals (As, Cd, Pb, Al, Cr, Zn, Co, Mo, Cu, Mn, V, B, Sr, Sb, Ti, Tl) in raspberry leaves from conventional, organic and wild crops and two years of cultivation are presented in Table 2.

**Table 2.** The content of selected heavy metals in raspberry leaves from organic, conventional and wild plants (mg kg$^{-1}$ dry matter).

| Heavy Metals | Cultivation | Conventional | | Organic | | Wild Crop | |
|---|---|---|---|---|---|---|---|
| | Year | 2020 | 2021 | 2020 | 2021 | 2020 | 2021 |
| Arsenic (As) | | 0.30 [d] ± 0.07 | 0.09 [a,b] ± 0.00 | 0.43 [c] ± 0.01 | 0.06 [a] ± 0.00 | 0.40 [c] ± 0.03 | 0.12 [b] ± 0.01 |
| Cadmium (Cd) | | 0.07 [a] ± 0.01 | 0.12 [a] ± 0.01 | 0.09 [a] ± 0.01 | 0.29 [b] ± 0.01 | 1.74 [d] ± 0.06 | 1.37 [c] ± 0.11 |
| Lead (Pb) | | 0.34 [a] ± 0.12 | 0.36 [a] ± 0.04 | 0.48 [a] ± 0.03 | 0.91 [b] ± 0.06 | 0.85 [b] ± 0.03 | 1.40 [c] ± 0.12 |
| Aluminium (Al) | | 69.68 [e] ± 1.07 | 108.95 [a] ± 4.07 | 50.41 [c] ± 2.46 | 38.43 [b] ± 2.42 | 59.51 [d] ± 4.93 | 108.62 [a] ± 0.52 |
| Chromium (Cr) | | 2.50 [b] ± 1.19 | 0.19 [a] ± 0.01 | 4.68 [d] ± 0.61 | 0.08 [a] ± 0.00 | 3.60 [b] ± 0.27 | 0.72 [a] ± 0.14 |
| Zinc (Zn) | | 48.89 [b,c] ± 0.58 | 65.30 [c] ± 8.50 | 26.25 [a] ± 1.68 | 35.99 [a,b] ± 0.21 | 138.08 [e] ± 22.11 | 94.67 [d] ± 6.43 |
| Cobalt (Co) | | 1.22 [b] ± 0.33 | 0.04 [a] ± 0.00 | 1.93 [d] ± 0.10 | 0.04 [a] ± 0.00 | 1.63 [c] ± 0.12 | 0.06 [a] ± 0.01 |
| Molybdenum (Mo) | | 3.01 [d] ± 0.44 | 3.84 [b] ± 0.26 | 1.99 [a] ± 0.08 | 0.56 [c] ± 0.05 | 1.62 [a] ± 0.09 | 4.25 [b] ± 0.31 |
| Copper (Cu) | | 6.51 [c] ± 0.15 | 5.71 [a] ± 0.44 | 6.07 [a] ± 0.05 | 5.17 [b] ± 0.05 | 6.99 [d] ± 0.28 | 6.03 [a] ± 0.14 |
| Manganese (Mn) | | 238.07 [d] ± 4.63 | 177.19 [c] ± 4.88 | 128.15 [b] ± 3.89 | 525.50 [a] ± 9.37 | 1631.64 [e] ± 29.31 | 547.71 [a] ± 16.78 |
| Vanadium (V) | | 0.18 [b] ± 0.03 | 0.36 [c] ± 0.02 | 0.14 [a,b] ± 0.01 | 0.13 [a] ± 0.00 | 0.13 [a] ± 0.01 | 0.75 [d] ± 0.03 |
| Boron (B) | | 75.38 [c] ± 8.81 | 117.02 [d] ± 3.73 | 33.87 [a] ± 4.23 | 48.01 [b] ± 0.69 | 39.82 [a] ± 2.29 | 48.20 [b] ± 0.73 |
| Strontium (Sr) | | 26.59 [a] ± 2.42 | 69.75 [c] ± 12.32 | 14.69 [b] ± 0.56 | 23.57 [a,b] ± 0.41 | 17.31 [a,b] ± 1.23 | 25.19 [a] ± 2.26 |
| Antimony (Sb) | | 0.24 [c] ± 0.04 | 0.04 [a,b] ± 0.01 | 0.33 [d] ± 0.01 | 0.02 [a] ± 0.00 | 0.36 [b] ± 0.03 | 0.27 [c] ± 0.01 |
| Titanium (Ti) | | 8.16 [a] ± 1.12 | 25.13 [e] ± 1.17 | 10.11 [b] ± 0.48 | 11.66 [c] ± 0.64 | 9.40 [a,b] ± 0.76 | 18.55 [d] ± 0.68 |
| Thallium (Tl) | | 0.0034 [a] ± 0.0017 | 0.0056 [b] ± 0.0002 | 0.0018 [a] ± 0.0001 | 0.0044 [ab] ± 0.0003 | 0.0108 [c] ± 0.0004 | 0.0235 [d] ± 0.0033 |

Results are presented as mean value ± standard deviation (n = 3). a, b, c . . . —means with different superscript letters in common differ significantly ($p < 0.05$).

The raspberry leaves from the investigated crops were characterised by various heavy metals contents. The results obtained showed that of the analysed samples, leaves of wild-growing raspberries had the highest amounts of cadmium, lead, zinc, manganese and vanadium.

The As content in the leaves was lower for all crops in the second harvest year, and these results are statistically significant compared to the previous harvest year. As for arsenic content, the results obtained by other authors were higher [28] compared to our findings.

The Cd content in wild-growing raspberry leaves was considerably higher than that of other crops. The raspberry leaves from the organic farming, harvested in 2021, also had more cadmium, by 211% compared to the previous year. This is congruent with the findings of other authors [41], who have also observed higher contents of this element. The levels of lead and cadmium determined by Biel et al. [42] were significantly higher than our results and were 5.76 and 2.91 mg kg$^{-1}$, respectively. In Angin et al. [43], the cadmium content was 1.44 mg kg$^{-1}$ DW in raspberry leaves (*Rubus idaeus* L.), which is a value similar to the value obtained in this study in wild raspberry leaves. Low results were obtained in the study of [44], which ranged within 0.0007–0.0029 mg kg$^{-1}$ DW.

Wild-growing and organic raspberry leaves harvested in 2021 contained much more lead, which was statistically significant—compared to specific crops and varieties. Compared to our results, the authors reported a higher Pb content in raspberry leaves (1.6 mg kg$^{-1}$) than the highest result obtained [28,41]. Other authors also obtained lower results [43,44].

As for zinc, its content in the leaves of wild-growing raspberry was significant. The content of zinc, determined in a control sample by Nejem et al. [33], was 43.69 mg kg$^{-1}$, which means that the result was close to that of conventional raspberries harvested in 2020. A higher content (90.3 mg kg$^{-1}$ DM) was reported by other authors [28], which was an approximate result for wild-growing raspberry from the 2021 harvest. On the

other hand, lower results were also noted [34,41,42,45]. Horuz et al. [46], who examined leaves of various *Rubus idaeus* L. raspberry varieties, determined Zn content at the level of 63.15–77.22 mg kg$^{-1}$, which corresponds to the result obtained for raspberry leaves from conventional farming conducted in 2021. In contrast, in Angin et al.'s [43] study, the content was 35.47 mg kg$^{-1}$ DW, which corresponds to the value of zinc content obtained in the second year of harvest in organic raspberry leaves. Similar results were obtained to the previous ones obtained in the study [44,47].

In the second year of harvest, cobalt contents in all crops were lower, which is consistent with the findings of other authors [41]. On the other hand, the Co content in raspberry leaves, determined by Biel et al. [42] was 0.42 mg kg$^{-1}$.

The examined raspberry leaves from the second harvest year had significantly lower Cr contents compared to those collected in 2020. The content of this element noted by Staszowska-Karkut [41] and Biel et al. [42] and was about 1 mg kg$^{-1}$.

The results obtained for molybdenum content in raspberry leaves were within the range given in another study, which was 4–21 mg kg$^{-1}$ DM [41]. The upper limit was also reported by Biel at al. [42].

The manganese content was different in all examined samples, and the highest was in the leaves of wild-growing raspberry harvested in 2020. The results reported by other authors were significantly lower than our findings [34,41–43,46]. A higher manganese content was also observed than in our samples, excluding the leaves of wild raspberry from the second year of cultivation [47]. Low pH, the presence of clay soils, the water–air imbalance as well as the activity of rhizosphere bacteria may cause the accumulation of manganese in the plant [48,49].

Copper content was similar and statistically significant for each raspberry crop with regard to each harvest year. Cu contents given in the available literature are lower [28,33,34,42] or higher [41,43,46,47] compared to our results. Similar observations referring to Cu content were reported by other authors [45]. The study by Chwil and Kostryco examined the three cultivars 'Glen Ample', 'Laszka' and 'Radziejowa'. The content of copper in each variety was higher than in the study [44].

As for boron, its content in leaves of wild-growing and organically cultivated raspberry was similar; however, it was different than that established for the leaves derived from conventional farming, for which its amounts in successive crops were 75.38 mg kg$^{-1}$ and 117.02 mg kg$^{-1}$. These values were statistically significant both among themselves and in comparison with other crops. The results obtained by other authors were either in the range of our results [41,47] or were lower [45,46].

No literature was found to confirm the contents of vanadium, strontium, antimony, titanium and thallium in raspberry leaves coming from different crops.

## 4. Conclusions

The occurrence of heavy metals is a major problem of environmental pollution and can have serious health consequences. It is commonly believed that organic food is healthier because is produced without the use of pesticides and other plant protection products.

Statistical analysis showed that significantly more ($p \leq 0.05$) cadmium, zinc, manganese and vanadium was found in the raspberry fruits from organic farming compared to those from conventional cultivation. In turn, fruits of wild-growing raspberry contained significantly more ($p \leq 0.05$) cadmium, zinc, copper and manganese compared to raspberry leaves and fruits from conventional and organic farming.

Wild-growing raspberry leaves had significantly more ($p \leq 0.05$) cadmium, lead, zinc, manganese and thallium than those originated from other crops. The raspberry leaves from organic cultivation contained statistically significant amounts of chromium, cadmium and lead compared to leaves from conventional cultivation. The latter however had higher levels of boron, strontium and titanium ($p \leq 0.05$) than leaves from other crops.

The leaves and fruits of wild-growing raspberry had significantly more heavy metals. Since the different heavy metals predominate in raspberries from conventional and organic

farming, additional research in the long term is needed to find out which type of raspberry cultivation is more beneficial for consumers due to the lower heavy metals content.

The article tried to give an answer to the potential consumer as to whether one should go for organic products, whether it is enough to buy fruit at the market (conventional) or one should buy them wild. Science is an ever-evolving field, so no definite answers can be given, but can only inform the consumer. Final conclusions should be drawn by potential consumers themselves.

**Author Contributions:** Conceptualization, M.K. and J.K.-D.; Methodology, S.S.; Investigation, S.S.; Writing—Original Draft Preparation, M.K.; Writing—Review and Editing, J.K.-D.; Supervision, J.K.-D.; Funding Acquisition, M.K. All authors have read and agreed to the published version of the manuscript.

**Funding:** This research received no external funding.

**Institutional Review Board Statement:** Not applicable.

**Informed Consent Statement:** Not applicable.

**Data Availability Statement:** Not applicable.

**Conflicts of Interest:** The authors declare no conflict of interest.

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
