# Peer review of "Evaluation of Selected Heavy Metals Contaminants in the Fruits and Leaves of Organic, Conventional and Wild Raspberry (Rubus idaeus L.)"

_applsci, doi:10.3390/app12157610_

Round 1

Author Response

Dear Reviewer,
all Your comments have been taken into account and their description can be found in the appendix, as well as in the manuscript text sent after revisions.
Thank You very much for reviewing the paper and all Your valuable comments.
Regards,
Joanna Kapusta-Duch

Reviewer 2 Report

The paper presents a comparison of the heavy metal content in organic, conventional and wild-grown raspberries. Given the health importance of these fruits, the findings should be of interest to the potential reader. I provide detailed comments below.

I encourage the use of the template, although it is not obligatory.

12, 74 analyze of raspberry > analyse heavy metals in raspberry (and the additional comment: (The authors are more likely to use the ending “sed” in the article - British English, so I think the verb form should be changed in this way.)

14, 124, 126, 149 analyzed > analysed (the same comment as for “analyse”)

15, 126 varies > varied

13 ICP-OES (Please explain the abbreviation.)

14 ICP-MS/MS (Please explain the abbreviation.)

14-23 (Please shorten the results and add conclusions.)

57 heave > heavy

65, 136, 144, 260 characterized > characterised (the same comment as for “analyse”)

74-80 (I suggest that the sentence “According...” be placed at the beginning of the paragraph and the next sentence start with the phrase: “Therefore, the aim...”.)

95 0.5 g > The 0.5 g

115, 117, 252, 254 Raspberry > raspberry

Table 1, Table 2 (I suggest that the names of the elements be written in capital letters.)

123-125 (This sentence is incomprehensible. Please rephrase it.)

142 DW (Please explain the abbreviation on first use.)

146 FW (Please explain the abbreviation on first use.)

145 content this > content of this

152 1st > first

172 Manganese > manganese

215-234 (These two paragraphs contain general information on heavy metals and should not be placed here but in the “Introduction” section. Moving them seems necessary, although unfortunately this will require renumbering the references.)

233 2,% (unintelligible notation)

Table 2 (Please do not split the table between pages.)

272 results. the > results the

273 1,6 > 1.6

281 correspond > corresponds

287 [36]. > [36],

290 by [37]. > by Biel at al. [37].

303 strontium. antymony > strontium, antimony

322 research is > research in the long term is

Author Response

(The authors gave the same response as above.)

Reviewer 3 Report

The manuscript entitled " Evaluation of selected heavy metals contaminants in the fruits and leaves of organic, conventional and wild raspberry (Rubus idaeus L.)" mainly focuses on analysis of heavy metals in the raspberry fruits and leaves. From the contents of the manuscript to estimate, this research is good in the qualitative and quantitative organization. In my judgement, it needs to be revised for publication in this journal.

General comments

Language of the manuscript must be revised carefully as there are many typos and grammatical mistakes in the sentences.

How the selected soil was examined in terms of the heavy metals, what are the sources of these heavy metals in the respective area where the raspberry grown or the heavy metals were introduced by the authors for experimental purpose.

  Abstract

Revise the sentence ‘analyze of raspberry fruits and leaves from organic”

Use full form of the abbreviations at first use.

Write down main methods in the abstract

Also add quantitative and main results in the abstract

Use consistent language as the authors used Cd in one place and chromium, cadmium and lead in other sentences.

Introduction

Line 35 -36 how these metals accumulation caused by anthropogenic activities. Write causes.

The author must discuss recent advances of the use of techniques for the removal of heavy metals.

Line 66-73 write economic and medicinal impacts of raspberry but also write yield losses and other effects of raspberry caused by heavy metals.  

Cite the sentence “expression of genes responsible for the transport of these substances in plants” with https://doi.org/10.3389/fgene.2021.635043.

How the authors have determined that the heavy metals are accumulated from two years in the wild plants?

Methodology

This section is well written.

Add sources of these contents in the land where the plants were cultivated.

Write the complete process of quantifying heavy metals in the plants fruit and leaves

Results

Section 3.1. why plants showed lower contents of heavy metals in the second year

Discussion is poorly written and mostly sentences are not cited and not related with the theme.

Conclusion must be revise, improve the English to clearly convey the meaning. Add future perspectives. What are the possibilities of research in future on this topic. 

Author Response

(The authors gave the same response as above.)

Round 2

Reviewer 1 Report

Accept in present form.

Reviewer 3 Report

I have no further comments